# Bioinspired trimesic acid anchored electrocatalysts with unique static and dynamic compatibility for enhanced water oxidation

Xiaojing Lin[1], Zhaojie Wang [1]✉, Shoufu Cao[1], Yuying Hu[1], Siyuan Liu [1]✉, Xiaodong Chen[1], Hongyu Chen[2], Xingheng Zhang[1], Shuxian Wei[2], Hui Xu[1], Zhi Cheng[1], Qi Hou[1], Daofeng Sun [1] & Xiaoqing Lu [1]✉

Layered double hydroxides are promising candidates for the electrocatalytic oxygen evolution reaction. Unfortunately, their catalytic kinetics and long-term stabilities are far from satisfactory compared to those of rare metals. Here, we investigate the durability of nickel-iron layered double hydroxides and show that ablation of the lamellar structure due to metal dissolution is the cause of the decreased stability. Inspired by the amino acid residues in photosystem II, we report a strategy using trimesic acid anchors to prepare the subsize nickel-iron layered double hydroxides with kinetics, activity and stability superior to those of commercial catalysts. Fundamental investigations through operando spectroscopy and theoretical calculations reveal that the superaerophobic surface facilitates prompt release of the generated $O_2$ bubbles, and protects the structure of the catalyst. Coupling between the metals and coordinated carboxylates via C–O–Fe bonding prevents dissolution of the metal species, which stabilizes the electronic structure by static coordination. In addition, the uncoordinated carboxylates formed by dynamic evolution during oxygen evolution reaction serve as proton ferries to accelerate the oxygen evolution reaction kinetics. This work offers a promising way to achieve breakthroughs in oxygen evolution reaction stability and dynamic performance by introducing functional ligands with static and dynamic compatibilities.

The oxygen evolution reaction (OER) is a crucial electrochemical process that serves as the foundation for multiple energy storage applications, including rechargeable metal-air batteries, nitrogen reduction, carbon dioxide reduction, and water electrolyzers[1,2]. Although state-of-the-art noble metal electrocatalysts, such as iridium (Ir)- and ruthenium (Ru)-based catalysts, have been developed as commercial benchmarks for practical OER

applications, their scarcities and exorbitant prices have sparked the search for cost-effective catalysts[3,4]. Electrodes based on Ni, Co, and Fe have shown great potential in OER but failed to meet strict industrial criteria for reactivity and stability, e.g., a lifespan of >1000 h at a current density of >1000 mA cm$^{-2}$. More efficient electrocatalysts are urgently needed to meet commercial requirements.

[1]School of Materials Science and Engineering, China University of Petroleum, Qingdao 266580, P. R. China. [2]College of Science, China University of Petroleum, Qingdao 266580, P. R. China. ✉e-mail: wangzhaojie@upc.edu.cn; lsy@upc.edu.cn; luxq@upc.edu.cn

2D transition metal compounds with edge-sharing octahedral $MO_6$ layers have garnered significant attention for use in the OER[5]. For example, nickel-iron layered double hydroxides (NiFe-LDH) exhibited superior performance in the oxygen evolution reaction (OER) due to their unique 2D lamellar structure and controllable electronic properties[6,7]. However, the minimum overpotential of the NiFe-LDH catalyst was limited by the scaling relationships among the binding energies of *O, *OH and *OOH (* indicates the adsorbed state) according to the adsorbate evolution mechanism[8]. Conventional approaches such as morphology regulation, vacancy steering and dopant modification were successful in improving the activity. However, the difficulty in breaking the bottleneck of dynamics remains the critical factor limiting the activity at high current densities. Even worse, the energetically favored metal segregation results in continuous leaching of transition metals during long-term operation, which largely hampers large-scale commercialization[9–12]. Thus, it is important to gain a deeper understanding of the active sites and structural features of the LDH to tailor improved kinetics and stability together.

Inspired by the $Mn_4Ca$ oxo cluster in photosystem II, mimicking of enzymes exhibiting specific binding was expected to increase the catalyst activity, kinetics and stability. In fact, it was confirmed that carboxylate ligands from the amino acid resides of protein backbones stabilize the metal clusters in homogeneous systems[13]. Whether direct or indirect coordination occurs, carboxylic acid groups with different numbers, positions or orientations performed their respective functions. In particular, as internal bases, they promoted proton transfer via concerted proton-electron transfer (CPET) processes[14–17]. Similar processes have been used to stabilize Ru in different ruthenium complexes and enhance the activity for water oxidation[18]. Hence, introducing ligands in homogeneous systems is an effective method for improving catalyst stability. However, the synergy between anchoring the carboxylates and the multi-metal active sites in heterogeneous catalysts has never been fully understood, which hinders the rational design of practical electrocatalysts.

Herein, we introduce a static and dynamic compatibility anchor strategy and report a subsize NiFe-LDH nanosheet catalyst modified with trimesic acid (SU-NiFe-LDH(TA)). Specifically, by introducing functional ligands into the NiFe-LDH with regulated electrodeposited ions, the resulting superhydrophilic and superaerophobic SU-NiFe-LDH(TA) catalyst exhibited higher electrocatalytic performance and stability at a large current ($1500\ mA\ cm^{-2}$) for 1300 h, outperforming those reported until now. The hybrid catalyst showed a higher stability for more than 800 h at current densities larger than $1000\ mA\ cm^{-2}$ under industrial conditions (60 °C and 6 M KOH). Operando experimental results and density functional theory (DFT) calculations revealed that coordination between the ligands and NiFe-LDH via C–O–Fe bonding stabilized the active metal centers. Moreover, the dynamic evolution of the functionalized ligand accelerated the OER kinetics by activating the O–H bonds in the reaction intermediates.

## Results and Discussion

Considerable effort has been devoted to improving the OER activity and stability of NiFe LDH[19–23]. As discussed in Fig. S1, rapid degradation of the NiFe-LDH catalysts under alkaline OER conditions was caused by metal dissolution. Currently, there are few effective methods for inhibiting metal dissolution at the atomic scale, particularly for enhancing the long-term stability under industrial conditions. Thus, we anchored the metal atoms at the atomic level by alternating the structures of LDH to inhibit the leaching of the transition metal hydroxides in alkaline media.

### Design and characterization

Based on the processes of photosystem II and CPET in homogeneous catalysis[18], carboxylate groups were used to stabilize the metal centers in the LDH. Electrochemical deposition provided both coordinated and uncoordinated carboxylate ligands through redox reactions occurring near the electrode. NiFe-LDH with trimesic acid ligands was prepared through direct electrochemical deposition on carbon paper and named NiFe-LDH(TA)@cp. $Ni(NO_3)_2 \cdot 6H_2O$ served as both the conditioning agent and the Ni source. $FeSO_4 \cdot 7H_2O$ was used as the Fe source, while trimesic acid afforded carboxylate groups for the crucial proton transfers and metal anchoring. In addition, the SU-NiFe-LDH(TA)@cp was prepared via a similar method except that ferric nitrate ($Fe(NO_3)_3 \cdot 9H_2O$) with different dissociation constants and redox properties was used to replace the $FeSO_4 \cdot 7H_2O$. The fabrication of SU-NiFe-LDH(TA)@cp is illustrated schematically in Fig. 1a. During the electrochemical deposition process, the pH changed near the working electrode, and the deprotonated trimesic acid carboxylates bonded with the metal centers[17].

The phase composition of the resulting catalyst was detected by X-ray diffraction (XRD). As shown in Fig. S3, the diffraction peaks were indexed to those of $Ni(OH)_2$ (PDF# 38-0715), consistent with the LDH[24]. The scarce differences in the XRD patterns indicated that the phase composition was well maintained after the introduction of the ligand. High-resolution transmission electron microscopy (HR-TEM) images were used to provide structural information. The multilayered lamellar structure was identified, and the lattice spacings of NiFe-LDH@cp, NiFe-LDH(TA)@cp and SU-NiFe-LDH(TA)@cp were 0.7 nm, which corresponded to the (003) planes of NiFe-LDH (Fig. S4). The selected area electron diffraction (SAED) pattern shown in Fig. S4c (inset) for SU-NiFe-LDH(TA)@cp displayed Debye-Scherrer rings for the (113), (015) and (101) planes of $Ni(OH)_2$. The micromorphologies of the samples were assessed with SEM. As shown in Fig. 1b, after the introduction of the trimesic acid ligands, NiFe-LDH(TA)@cp exhibited a typical morphology with ultrathin wrinkled nanosheets. Unlike NiFe-LDH(TA)@cp, the SEM images shown in Fig. 1c and Fig. S5 for SU-NiFe-LDH(TA)@cp exhibited ultrathin nanosheets with smaller lateral sizes (~47.90 nm) and rougher surface structures. In Fig. S6, the TEM images of SU-NiFe-LDH(TA)@cp and NiFe-LDH@cp reveal obvious nanosheet structures. In contrast, there were more folds in SU-NiFe-LDH(TA)@cp. These results indicate that trimesic acid bonded to metal ions with different valence states during electrochemical deposition, and this affected the growth and aging of the LDH phase[17,25,26]. Among the metal ions, $Fe^{3+}$ induced the formation of smaller nanosheets due to the smaller acidity coefficient of $Fe^{3+}$ compared with that of $Fe^{2+}$ ($pKa[Fe^{3+}]$ =2.84, $pKa[Fe^{2+}]$=6.74) (Fig. S7), which was confirmed by the TEM images of SU-NiFe-LDH(TA)@cp and NiFe-LDH(TA)@cp presented in Fig. S8. It was evident that SU-NiFe-LDH(TA)@cp consisted of smaller nanosheets (~56.68 nm) than NiFe-LDH(TA)@cp (~733.67 nm), which was consistent with the SEM results. This provides an important strategy for optimizing the morphology to facilitate mass transfer and bubble diffusion.

In an effort to identify the composition of trimesic acid in the as-prepared samples, Fourier transform infrared (FTIR) spectra were obtained and showed in Fig. S9. Two distinct peaks at 1650 and $1382\ cm^{-1}$ were attributed to the bending vibrations of $H_2O$ and the stretching vibrations of $NO_3$ groups, respectively. Signals associated with M–OH bonds in the layered double hydroxides appeared within the range $400–650\ cm^{-1}$. Compared with NiFe-LDH, TA and TANa, the peaks at 1620 and $1433\ cm^{-1}$ were ascribed to $\upsilon_{as}$(–COO–) and $\upsilon_s$(–COO–). The broad bands at 769 and $727\ cm^{-1}$ were related to vibrations of the benzene rings in TA. Significantly, two peaks at 1160 and $850\ cm^{-1}$ were observed for the SU-NiFe-LDH(TA) sample, which corresponded to C–O–Fe vibrational modes. These results indicated that the trimesic acid ligands were successfully anchored in NiFe-LDH. The Raman spectra of the NiFe-LDH@cp and NiFe-LDH(TA)@cp electrodes (Fig. S10) contained two intense signals at 455 and $546\ cm^{-1}$, which were assigned to the $E_g$ bending mode and $A_{1g}$ stretching mode of M–OH species in the LDH, e.g., Ni–O signals and overlapped Fe–O signals[20,27]. Notably, the Ni–OH peaks for SU-NiFe-LDH(TA)@cp were

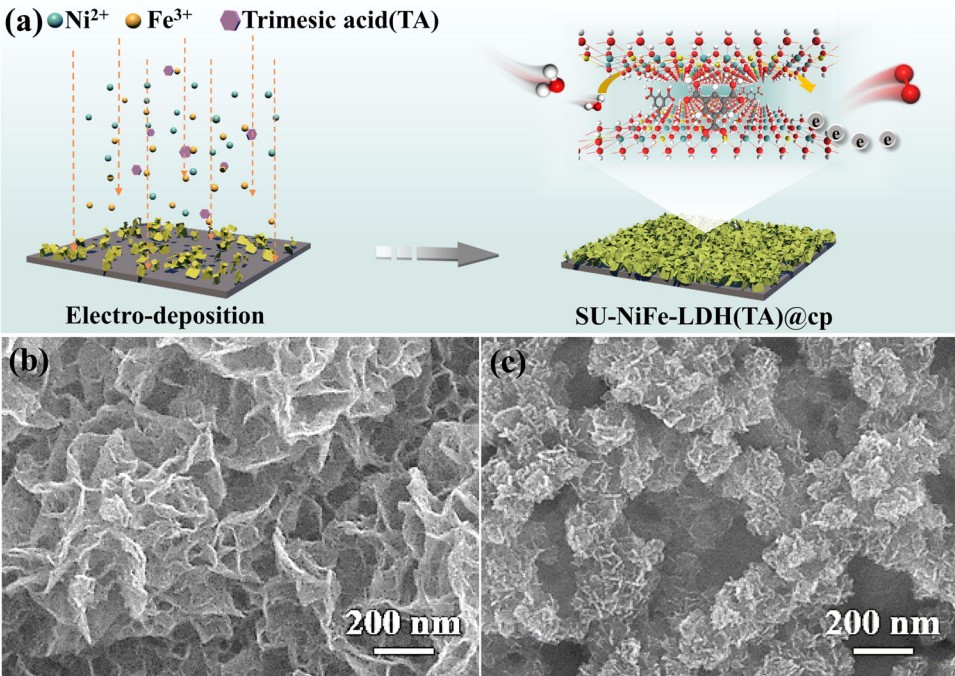

**Fig. 1 | Schematic of the electrodeposition and structure characterization. a** Schematic illustration of the preparation procedure for SU-NiFe-LDH(TA)@cp. **b, c** SEM images of NiFe-LDH(TA)@cp and SU-NiFe-LDH(TA)@cp.

redshifted to 448 and 530 cm$^{-1}$, respectively. This indicated electron transfer between Ni and Fe[20,28]. Two peaks at 683 and 721 cm$^{-1}$ for SU-NiFe-LDH(TA)@cp and NiFe-LDH(TA)@cp were ascribed to the M–OH signals of the LDH and C–H signals of trimesic acid, respectively. The stretching modes of coordinated carboxylate groups derived from trimesic acid were observed at 1421 cm$^{-1}$. These characterization data suggested that trimesic acid was successfully anchored in the NiFe-LDH.

The electronic states of the samples were investigated with X-ray photoelectron spectroscopy (XPS). As shown in Fig. 2a and Fig. S11, the Ni 2$p$ peaks for NiFe-LDH@cp and NiFe-LDH(TA)@cp located at 856.1 eV and 873.7 eV were ascribed to Ni 2$p_{3/2}$ and Ni 2$p_{1/2}$ binding energies, respectively. The Ni 2$p_{3/2}$ and Ni 2$p_{1/2}$ peaks for SU-NiFe-LDH(TA)@cp appeared at 855.5 and 873.1 eV, respectively. The shifts originated from the different dissociation constants and valence states of the metal ions (Fe$^{2+}$/Fe$^{3+}$ salts), which slightly modified the local electronic structure of the Ni atom. In Fig. 2b, the Fe 2$p_{3/2}$ and Fe 2$p_{1/2}$ peaks of SU-NiFe-LDH(TA)@cp were broader than those of NiFe-LDH@cp and were deconvoluted into six peaks. The peaks at 712.5 and 725.5 eV, along with the satellite peaks, confirmed the presence of Fe$^{3+}$, which was consistent with NiFe-LDH@cp. The peaks at 708.6 and 722.3 eV were assigned to C–O–Fe species, which proved that trimesic acid was anchored to the LDH by C–O–Fe bonds[29,30]. The high-resolution C 1$s$ spectra for SU-NiFe-LDH(TA)@cp and NiFe-LDH(TA)@cp were deconvoluted into peaks for three surface carbon components (Fig. 2c), which were nonoxygenated carbon (C–C: 284.8 eV), oxygen-containing carbon (C–O: 286.2 eV), and carboxyl carbon (O = C–O: 288.1 eV) in the ligands[31]. Additionally, three oxygen species, including adsorbed water molecules (Ow), defects with low oxygen coordination levels (Ov), and metal–OH bonds (M–OH), showed peaks at 529.0, 530.5, and 536.2 eV in Fig. 2d and Fig. S11d, respectively[6,32]. Additionally, the proportion of Ov in the SU-NiFe-LDH(TA)@cp sample was dramatically higher (26.43%) than those in NiFe-LDH(TA)@cp (16.34%) and NiFe-LDH@cp (16.61%). These results indicated that ligand functionalization and structure optimization resulted in smaller and rougher nanosheet structures, which

provided more edge sites and optimized the local electronic structures.

### Enhanced OER electrocatalytic activity and stability

To demonstrate the use of ligand functionalization and structural optimization to enhance the electrocatalytic performance, a typical three-electrode system immersed in a 1 M KOH electrolyte solution was used to study the electrocatalytic OER performance of these LDH samples (Fig. 3). Linear sweep voltammograms (LSV) were measured with scan rates of 5 mV s$^{-1}$ with 90% iR compensation unless otherwise stated. Notably, the SU-NiFe-LDH(TA)@cp catalyst displayed the highest OER catalytic activity (Fig. 3a). According to the LSV curves, it exhibited the lowest overpotential of 248 mV at a current density of 100 mA cm$^{-2}$ and outperformed NiFe-LDH(TA)@cp (265 mV) and NiFe-LDH@cp (308 mV). Figure 3b shows the Tafel plots for the catalyst samples, and both SU-NiFe-LDH(TA)@cp and NiFe-LDH(TA)@cp exhibited lower Tafel slopes of 31.1 and 40.6 mV dec$^{-1}$, respectively, compared to that of NiFe-LDH@cp (98.4 mV dec$^{-1}$). The results indicated accelerated kinetics of the catalyst after carboxyl functionalization. The activities and kinetics of the catalysts prepared via ligand introduction were better than those of previously reported LDH-based catalysts (Fig. 3c and Table S1). Electrochemical impedance spectroscopy (EIS) was used to study the catalytic kinetics[33]. As shown in Fig. 3d and Table S2, SU-NiFe-LDH(TA)@cp exhibited the smallest semicircle radius (1.69 Ω cm$^{-2}$) and the smallest charge-transfer resistance, implying fast charge-transfer kinetics in the OER. The electrochemically active surface area (ECSA) of the catalyst was calculated from the double-layer capacitance ($C_{dl}$) (Fig. S12). The $C_{dl}$ was calculated from cyclic voltammetry (CV) curves obtained with different scan speeds over the potential range 1.183 to 1.233 V vs. RHE. As seen in Fig. 3e, SU-NiFe-LDH(TA)@cp exhibited the largest $C_{dl}$ of 3.2 mF cm$^{-2}$, which was nearly twice that of NiFe-LDH(TA)@cp (1.64 mF cm$^{-2}$) and four times that of NiFe-LDH@cp (0.81 mF cm$^{-2}$). The higher $C_{dl}$ was mainly due to the unique morphology of SU-NiFe-LDH(TA)@cp, which provided many exposed active sites. To compare the intrinsic activities, the current densities were normalized with the electrochemical

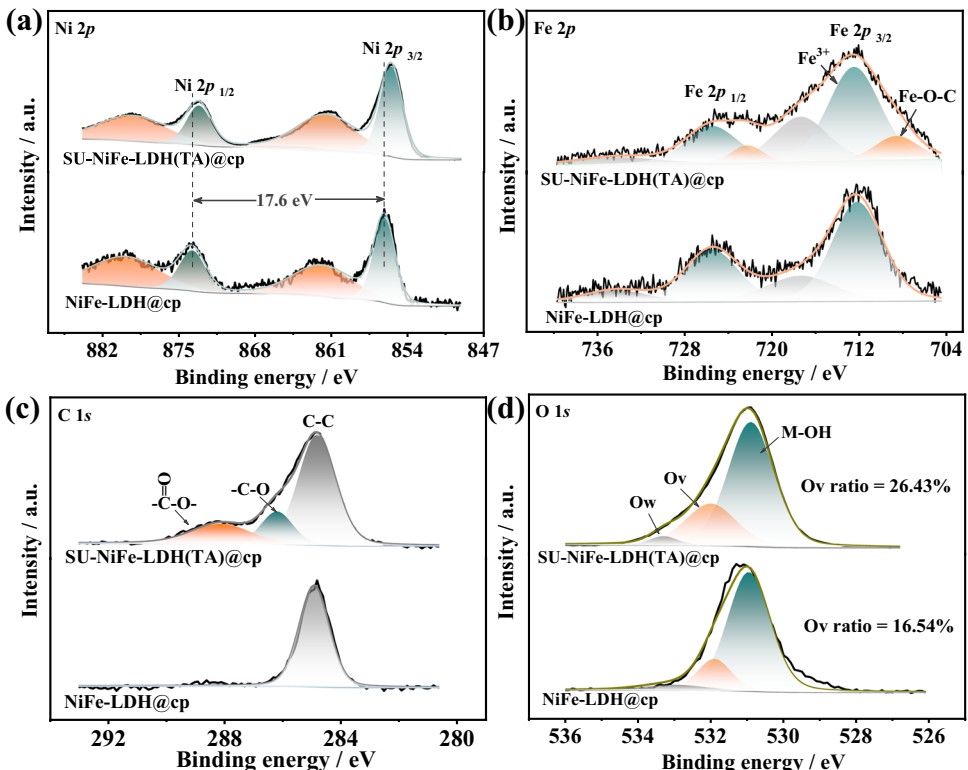

**Fig. 2 | XPS measurements.** High-resolution XPS spectra of (**a**) Ni 2*p*, (**b**) Fe 2*p*, (**c**) C 1*s*, and (**d**) O 1*s* for NiFe-LDH@cp and SU-NiFe-LDH(TA)@cp electrodes.

surface areas of the catalysts (Fig. S13a). As a result, the SU-NiFe-LDH(TA)@cp catalyst showed the highest intrinsic activity among the samples, indicating that both the trimesic acid ligands and the improved active surface area contributed to the increased activity. The LSV curves for the as-prepared samples were shown in Fig. S13b without iR correction. The SU-NiFe-LDH(TA)@cp catalyst also showed optimal catalytic performance. Stability is another critical factor for OER electrocatalysts. The carboxyl-functionalized catalysts exhibited significantly improved stabilities. For example, NiFe-LDH(TA)@cp showed a higher stability after OER catalysis for 24 h (Fig. S14). The chronoamperometry curves for SU-NiFe-LDH(TA)@cp (Fig. S15) were determined with 1 M KOH. Degradation of the current density was negligible after continuous testing for 150 h at 1.73 V vs. RHE in a three-electrode system. XRD, FTIR and Raman analyses were conducted on SU-NiFe-LDH(TA)@cp after OER stability testing (p-SU-NiFe-LDH(TA)@cp). As shown in Fig. S16, the XRD pattern for p-SU-NiFe-LDH(TA)@cp was consistent with that of fresh SU-NiFe-LDH(TA). In Fig. S17, the Raman peaks for p-SU-NiFe-LDH(TA)@cp shifted to 474 and 544 $cm^{-1}$, which were associated with Ni–O vibrations in NiOOH moieties. In addition, Raman peaks for coordinated and uncoordinated carboxylates were observed in p-SU-NiFe-LDH(TA)@cp, indicating that the trimesic acid ligands were stably anchored to the NiFe-LDH during the OER stability tests. This was validated by the FTIR spectrum of p-SU-NiFe-LDH(TA)@cp. As seen in Fig. S18, the observed peak at 573 $cm^{-1}$ was attributed to M–O. The signals associated with coordinated carboxylates were detected at 1373, 769 and 727 $cm^{-1}$. The signals associated with uncoordinated carboxylates appeared at 1608, 1454, 1404, 1276, 1246, 744 and 688 $cm^{-1}$. In addition, the vibrational peaks of Fe–O–C were observed at 1160 and 850 $cm^{-1}$ in the p-SU-NiFe-LDH(TA)@cp spectrum, which confirmed the higher structural stability. To investigate the impact of the carboxylate ligands on the electronic structure before and after the OER, additional analysis was needed. XPS was used to investigate the electronic states of the three samples after the stability test (p-NiFe-LDH@cp, p-NiFe-LDH(TA)@cp and p-SU-NiFe-LDH(TA)@cp), as shown in Fig. S19 and Table S3. In particular, the

peaks at 856.1 (Ni $2p_{3/2}$) and 873.7 eV (Ni $2p_{1/2}$) for NiFe-LDH@cp increased to 856.8 and 874.4 eV (ΔE = 0.7 eV) after the OER test (p-NiFe-LDH@cp). The Fe 2*p* peaks of p-NiFe-LDH@cp were located at 712.8 and 725.8 eV, which are 0.3 eV higher than those for NiFe-LDH@cp. Meanwhile, the proportion of $O_v$ for p-NiFe-LDH@cp increased to 53.23%, indicating that metal dissolution created more defects. However, few changes were observed in the XPS spectra of NiFe-LDH(TA)@cp and SU-NiFe-LDH(TA)@cp after the OER tests (named p-NiFe-LDH(TA)@cp and p-SU-NiFe-LDH(TA)@cp, respectively). These results suggested that the strategy of anchoring carboxylate ligands stabilized the structure and coordination environment of the catalyst, thereby enhancing its stability. In addition, studies of the active sites are particularly important for improving the stability. As shown in Fig. S20, Ni provided the main active sites, and TA anchoring of the Fe species was essential for high stability, as shown by comparing the activities of the samples with different metal and ligand contents. The concentration of dissolved metal ions was measured by ICP–MS (Fig. 3f and Table S4). Compared with NiFe-LDH@cp, the concentrations of Ni and Fe in the electrolyte were dramatically decreased for NiFe-LDH(TA)@cp and SU-NiFe-LDH(TA)@cp after 24 h of electrolysis. These results confirmed that the trimesic acid ligands stabilized the metal sites in NiFe-LDH, inhibited the dissolution of the metal ions and thus prolonged the lifetimes of the LDH electrocatalysts.

To enable application as an OER catalyst in different fields and with different equipment, SU-NiFe-LDH(TA) was prepared via electrochemical deposition on different conductive substrates, including carbon paper (cp), carbon cloth (cc), and nickel foam (nf) (Fig. S21a). Figure S21b shows the linear sweep voltammograms (LSV) of SU-NiFe-LDH(TA)@cc, SU-NiFe-LDH(TA)@nf and SU-NiFe-LDH(TA)@cp. The nearly overlapping curves suggested similar OER activities, demonstrating that the preparation of SU-NiFe-LDH(TA) met the requirements of different scenarios. Moreover, the performance of the targeted catalyst at an industrial current density (e.g., > 1000 mA $cm^{-2}$) was studied for practical application. An SU-NiFe-LDH(TA)@nf OER

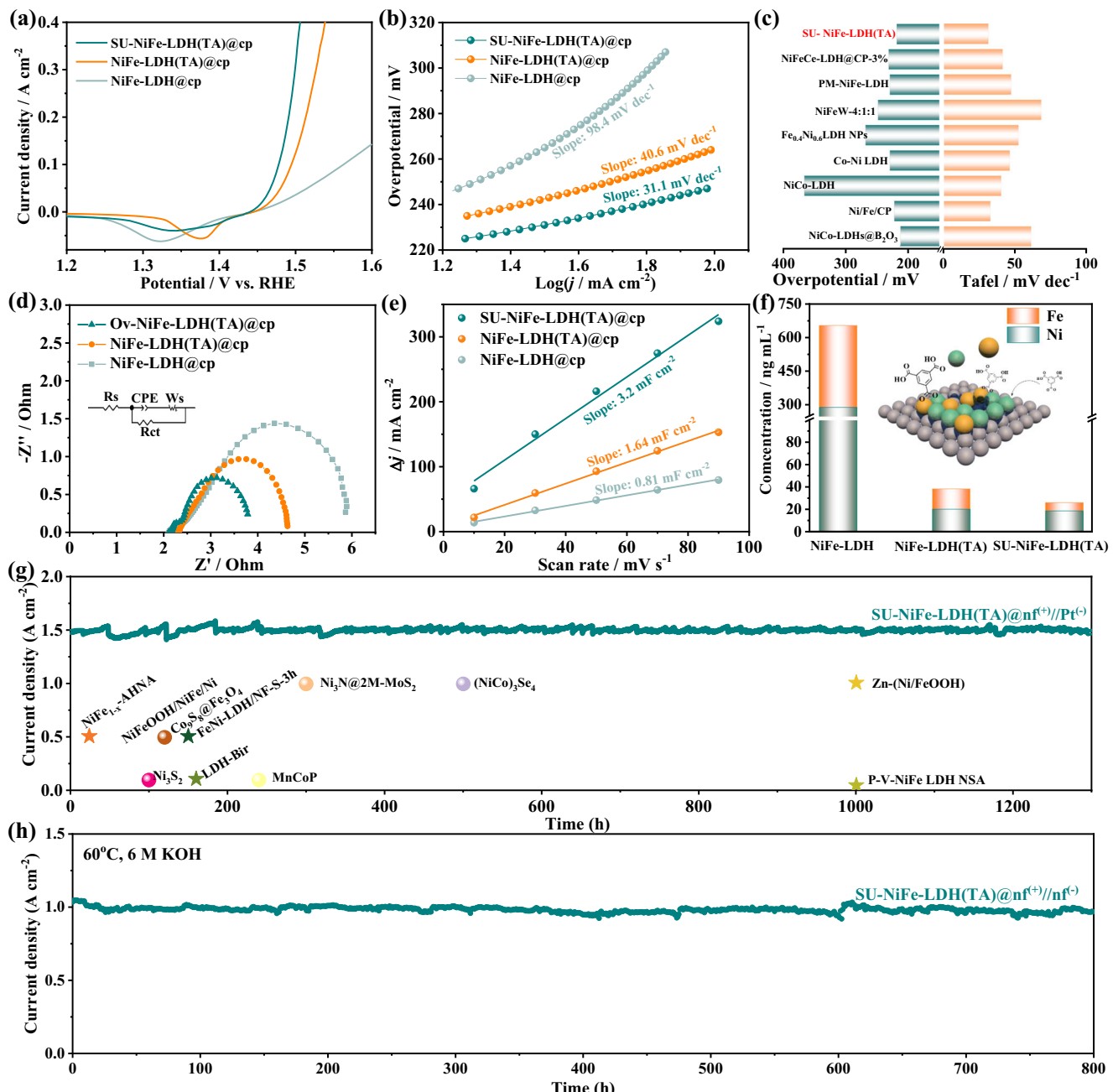

**Fig. 3 | Electrocatalytic performance of prepared electrodes. a** Linear sweep voltammetry with iR compensation, and **b** the corresponding Tafel plots of NiFe-LDH@cp, NiFe-LDH(TA)@cp, and SU-NiFe-LDH(TA)@cp. **c** Comparison of the current densities at 100 mV and Tafel plots among different catalysts in 1 M KOH. **d** Nyquist plots and the corresponding equivalent circuit. The solution resistance (Rs) of NiFe LDH@cp, NiFe-LDH(TA)@cp, and SU-NiFe-LDH(TA)@cp is 2.08, 2.27 and 2.12 Ω cm⁻², respectively. **e** Double-layer capacitances (C_dl). **f** Dissolved nickel and iron ions of NiFe LDH@cp, NiFe-LDH(TA)@cp, and SU-NiFe-LDH(TA)@cp detected by ICP-MS after electrolysis for 24 h in 1 M KOH. **g** Chronoamperometric curves of SU-NiFe-LDH(TA)@nf and comparison among different catalysts in 1 M KOH. **h** Chronoamperometric curves of SU-NiFe-LDH(TA)@nf in 6 M KOH at 60 °C.

electrode and a platinum HER electrode were paired to construct an alkaline water electrolysis system (SU-NiFe-LDH(TA)@nf$^{(+)}$//Pt$^{(-)}$AWE). As presented in Fig. 3g, the SU-NiFe-LDH(TA)@nf$^{(+)}$//Pt$^{(-)}$AWE system maintained approximately 98% of the initial current density (1500 mA cm⁻²) after 1300 h of continuous operation at 2.18 V. The stability was superior to those of recently reported advanced catalysts (Table S5 and S6). The SEM images of SU-NiFe-LDH(TA)@nf before and after continuous operation (1300 h) exhibited similar lateral sizes (~ 44.98 nm) and ultrathin nanosheet morphologies (Fig. S22), which corroborated the higher structural stability of SU-NiFe-LDH(TA) during industrial use. To move further toward industrial application, an SU-

NiFe-LDH(TA)@nf OER electrode and a commercial nickel foam (nf) HER electrode were paired to construct an alkaline water electrolysis cell (SU-NiFe-LDH(TA)@nf$^{(+)}$//nf$^{(-)}$AWE) for use under industrial conditions (60 °C and 6 M KOH). As shown in Fig. 3h, the SU-NiFe-LDH(TA)@nf$^{(+)}$//nf$^{(-)}$AWE cell delivered a current density of 1000 mA cm⁻² at 1.90 V and maintained its activity without decay for more than 800 h.

## Fundamental origins of the enhanced catalytic activity and stability

As discussed above, structure optimization and ligand functionalization inhibited metal leaching and dissolution and thus improved the

activity and stability of SU-NiFe-LDH(TA). The corresponding evolution law and mechanism for the OER process could provide insight into the design and development of electrocatalysts.

Contact angle tests with water droplets and studies of the release mechanism for $O_2$ bubbles are critical for understanding mass transfer and the stability of the OER electrode[34]. Fig. 4a–c displays the contact angle test results for NiFe-LDH@cp, NiFe-LDH(TA)@cp, and SU-NiFe-LDH(TA)@cp. The carbon paper used in this study was hydrophobic (Fig. S23). The contact angles for ultrapure water on NiFe-LDH@cp and NiFe-LDH(TA)@cp were 18° and 15°, respectively, while the contact angle of ultrapure water on SU-NiFe-LDH(TA)@cp was close to 0°. The rough surface of SU-NiFe-LDH(TA)@cp with the smaller nanosheets provided superhydrophilicity. Interestingly, the contact angle for ultrapure water on NiFe-LDH(TA)@cp dropped sharply to 0° after standing for 0.1 s but remained at 14° on NiFe-LDH@cp (Fig. S24). These significant changes showed that the inserted layers formed by trimesic acid increased the rate of electrolyte diffusion across the surfaces and between the layers. In addition, since the cavitation effect caused by the bursting bubbles could cause serious damage to the nanostructure[35,36], the evolution of bubbles on the electrode surface could impact the structural stability[37]. Digital photographs of $O_2$ release from different electrodes were captured during galvanostatic scans at 200 mA cm$^{-2}$. As shown in Fig. S25, the oxygen bubbles generated on the NiFe-LDH(TA)@cp and SU-NiFe-LDH(TA)@cp electrodes were much smaller and almost invisible. To gain insight into the generation and disengagement of the bubbles, the process was recorded in situ with a 3D confocal microscope coupled with an electrochemical workstation. Figure 4d–f illustrates the formation and disengagement of $O_2$ bubbles over 40 s. The $O_2$ bubbles were firmly adhered to the surface of the NiFe-LDH@cp electrode and grew into a large bubble measuring 34.67 μm. In contrast, the $O_2$ bubbles were much smaller before escaping from NiFe-LDH(TA)@cp and SU-NiFe-LDH(TA)@cp. In particular, the diameter of the oxygen bubble on SU-NiFe-LDH(TA)

@cp was only 8.54 μm. This meant that the oxygen bubble disengaged immediately after it was formed, which provided much faster re-exposure of the catalytic sites to the surrounding electrolyte. The superaerophobic and superhydrophilic behavior of SU-NiFe-LDH(TA)@cp accelerated the release of the bubbles and mass transfer, which improved the activity and stability.

To understand the intrinsic mechanism for the efficient OER catalysis by SU-NiFe-LDH(TA)@cp, in situ Raman spectroscopy was used to monitor the evolution of the carboxyl groups in the trimesic acid ligands during water oxidation. In situ Raman spectra were collected over the potential range 0.30–0.58 V (vs. Ag/AgCl) with intervals of 20 mV. As shown in Fig. 5a and Fig. S26, the relative changes seen in the Ni–OH vibrations at 450 and 521 cm$^{-1}$ were related to oxidation during the OER process. Compared with NiFe-LDH@cp, weak peaks appeared at 1421 cm$^{-1}$ for the coordinated carboxylates (–COO–) in the NiFe-LDH(TA)@cp and SU-NiFe-LDH(TA)@cp samples, and the intensities varied little with the working potential. The coordinated carboxylates stabilized the metal centers and acted as static rivets to inhibit metal dissolution. Remarkably, as the potential was increased, a broad bond appeared at 1642 cm$^{-1}$ and was assigned to a stretching mode of the uncoordinated carboxyl groups (–COOH), and it remained stable during water electrolysis with the NiFe-LDH(TA)@cp and SU-NiFe-LDH(TA)@cp samples. In addition, in situ FTIR tests were used to study the dynamic evolution between uncoordinated (–COOH) and coordinated (–COO–) carboxylate groups and determine the electrochemical mechanism. The test pattern and details are provided in the Supporting Information and Fig. S27. The results are illustrated in Fig. 5b. In addition to the well-defined peak at 702 cm$^{-1}$, which was attributed to vibrations of the M–OH groups, two peaks appeared at 1620 cm$^{-1}$ ($v_{as}$(–COO–)) and 1433 cm$^{-1}$ ($v_s$(–COO–)) for the pristine SU-NiFe-LDH(TA)@nf at 0 V. With increasing voltage, two broad peaks appeared at approximately 3000 cm$^{-1}$ and were assigned to $v$(–OH) of the uncoordinated carboxylates (–COOH), and these remained stable

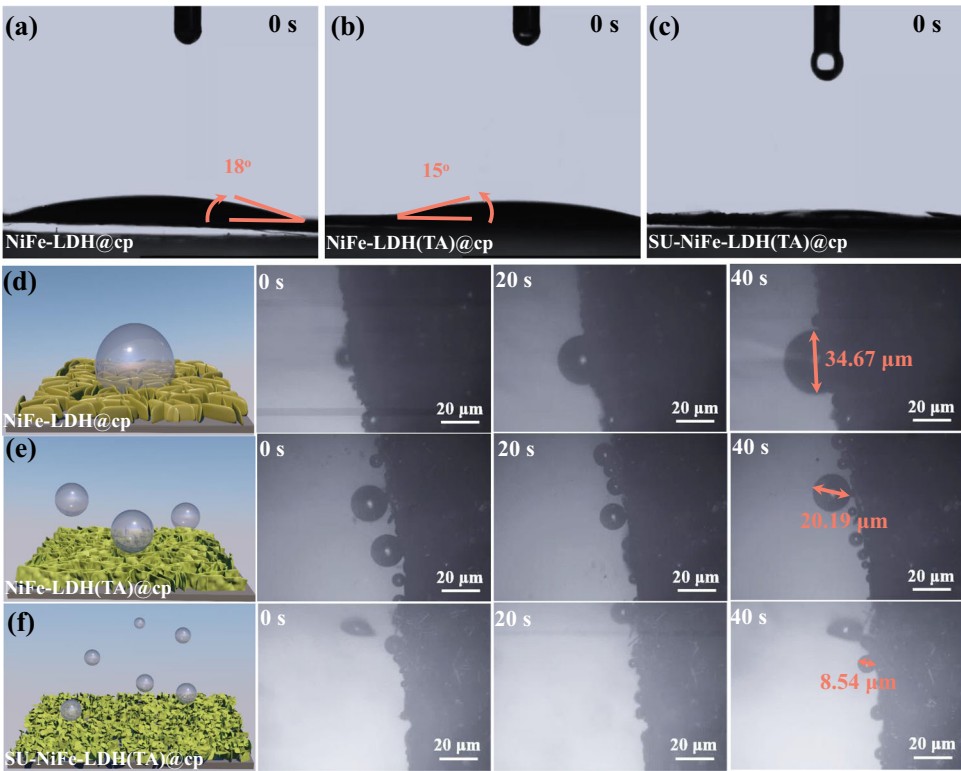

**Fig. 4 | Production and detection of in situ adhesion behavior. a–c** The contact angle of ultrapure water on NiFe LDH@cp, NiFe-LDH(TA)@cp, and SU-NiFe-LDH(TA)@cp. **d–f** In situ detection of precipitation behavior of $O_2$ on NiFe LDH@cp, NiFe-LDH(TA)@cp, and SU-NiFe-LDH(TA)@cp at 10 mA cm$^{-2}$, respectively.

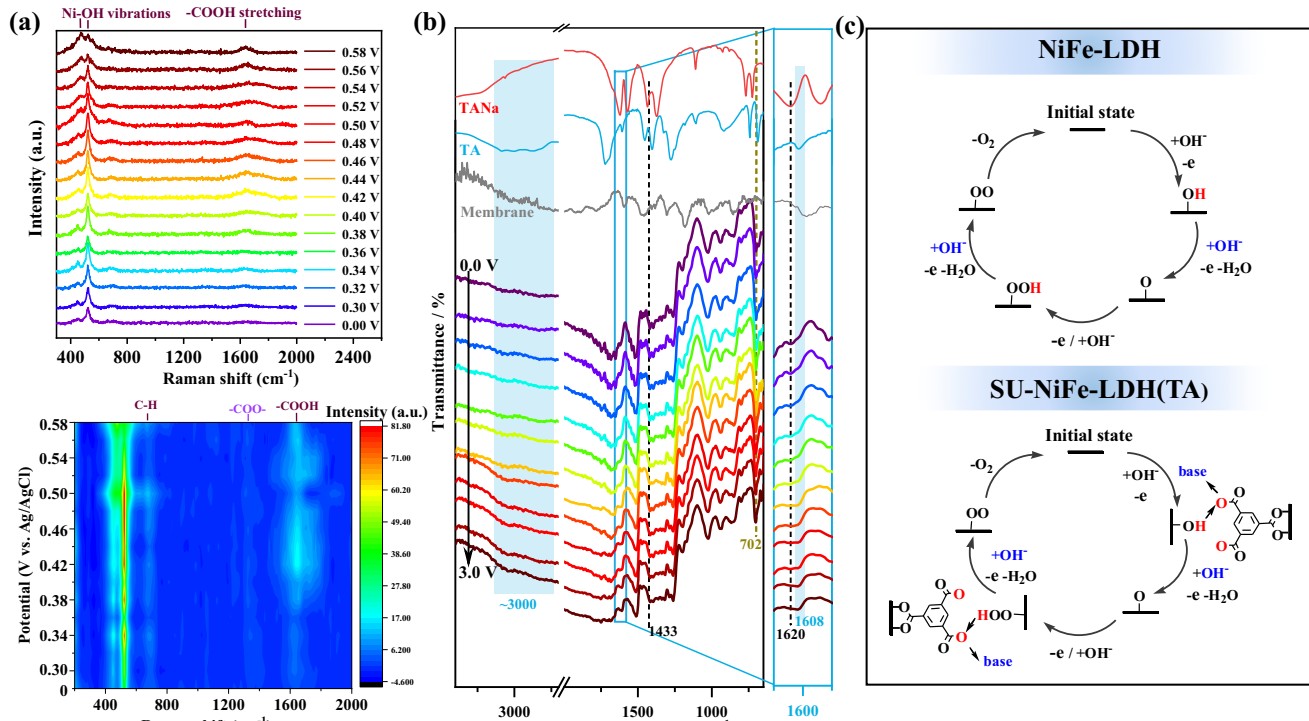

**Fig. 5 | The in situ dynamic evolution of carboxyl ligand. a** In situ Raman spectroscopy measurements on SU-NiFe-LDH(TA)@cp sample at different applied potentials (V vs. Ag/AgCl). **b** In situ FTIR spectroscopy measurements on SU-NiFe-LDH(TA)@cp sample at different applied potentials. **c** Proposed OER pathways with intermediates for NiFe-LDH and SU-NiFe-LDH(TA).

during the operation. Meanwhile, a peak centered at 1608 cm$^{-1}$ arose, and it corresponded to an antisymmetric stretching vibration of –COOH ($\upsilon_{as}$(–COOH)). These results provide solid evidence for the involvement of the carboxyl groups in proton transfer by the OER intermediates. Benefiting from the multiple carboxyl groups and their distribution in the trimesic acid molecules, the uncoordinated carboxylates located near the catalytic centers functioned as ferries to promote proton transfer from intermediate to base (Fig. 5c).

DFT calculations were conducted to investigate the mechanism for stabilization/activation of the metal active sites by the ligands. Two lamellar structures for pristine NiFe-LDH and SU-NiFe-LDH(TA) are illustrated in Fig. S28 and Fig. 6a. Anchoring of the metal centers by the trimesic acid in SU-NiFe-LDH(TA) was studied by the density of states (DOS) and charge analyses. The coordinated locations were labeled as SU-NiFe-LDH(TA)/Fe$_a$, SU-NiFe-LDH(TA)/Ni$_a$, SU-NiFe-LDH(TA)/O$_a$, and SU-NiFe-LDH(TA)/O$_b$. As seen in Fig. 6b, the 3d orbitals of SU-NiFe-LDH(TA)/Fe$_a$ overlapped more and thus had stronger interactions with the 2p orbitals of SU-NiFe-LDH(TA)/O$_a$, which facilitated the binding of the ligands to the metal centers in SU-NiFe-LDH(TA). The charge density difference in Fig. 6c indicated electrons were transferred from C (trimesic acid) and Fe to the neighboring O atom via C–O–Fe bonds. Bader charge analysis was used to explore charge transfer between the NiFe-LDH and the deprotonated trimesic acid ligands in SU-NiFe-LDH(TA). According to the Bader charges shown in Fig. S29, the electrons from Fe$_a$ and C (trimesic acid) were transferred to O$_a$ sites. The strong electronic interactions between the ligands and nickel-iron hydroxides in SU-NiFe-LDH(TA) stabilized the metal species and improved the stability of the SU-NiFe-LDH(TA) electrode during the OER.

Based on the anchoring structures, the Gibbs free energies for the neighboring Ni$_b$ and Fe$_b$ sites in SU-NiFe-LDH(TA) were studied and compared with those in NiFe-LDH (Figs. S30, S31 and S32). Figure 6d and Fig. S33 show the free energy profiles obtained for the OER at the

SU-NiFe-LDH(TA)/Ni$_b$, SU-NiFe-LDH(TA)/Fe$_b$, NiFe-LDH/Ni$_b$, and NiFe-LDH/Fe$_b$ sites. The rate-determining step (RDS) for the OER on the Ni$_b$ and Fe$_b$ sites in NiFe-LDH was the formation of *O and *OOH, respectively, and the highest free energy changes were 2.67 and 2.12 eV at U = 0 V. After introducing the ligands, the RDS was deprotonation of OH$^-$ to form *O in SU-NiFe-LDH(TA)/Ni$_b$. The reaction energies decreased to 2.06 and 1.94 eV at U = 0 V for SU-NiFe-LDH(TA)/Ni$_b$ and SU-NiFe-LDH(TA)/Fe$_b$, respectively, which were lower than those of NiFe-LDH/Ni$_b$ and NiFe-LDH/Fe$_b$. Specifically, the reaction energies for conversion of *OOH to O$_2$ were only 0.178 and 0.003 eV at U = 0 V for SU-NiFe-LDH(TA)/Ni$_b$ and SU-NiFe-LDH(TA)/Fe$_b$, respectively, which were much lower than those of NiFe-LDH/Ni$_b$ (0.470 eV) and NiFe-LDH/Fe$_b$ (1.154 eV). These results indicated that the uncoordinated carboxylates of SU-NiFe-LDH(TA) reduced the energy of the deprotonation step occurring during the OER process, which was related to the dynamic evolution of the uncoordinated carboxylates (Fig. 5). To understand the dynamic role of the uncoordinated carboxylates, the O–H bond lengths in both the *OOH and *OH intermediates were investigated. Figure 6e and Table S7 show that the O–H bond lengths in both the *OOH and *OH intermediates for SU-NiFe-LDH(TA) were longer than those in NiFe-LDH. This indicated that the uncoordinated carboxylates in the ligand acted as proton ferries to activate the O–H bonds in the intermediates and improve the kinetic performance via rapid proton transfer. The in situ Raman spectra and DFT calculations provided experimental and theoretical evidence for proton transfer via the uncoordinated carboxylates.

In conclusion, we report here NiFe-LDH catalysts modified with trimesic acid (SU-NiFe-LDH(TA)) that were grown on various conductive substrates. The small lateral sizes and ultrathin structures generated the superhydrophilic and superaerophobic character of the SU-NiFe-LDH(TA). It exhibited OER performance with a lower overpotential (219 mV at 10 mA cm$^{-2}$) and a decreased Tafel slope (31.1 mV dec$^{-1}$). Moreover, the material showed OER stability for over 1300 h at

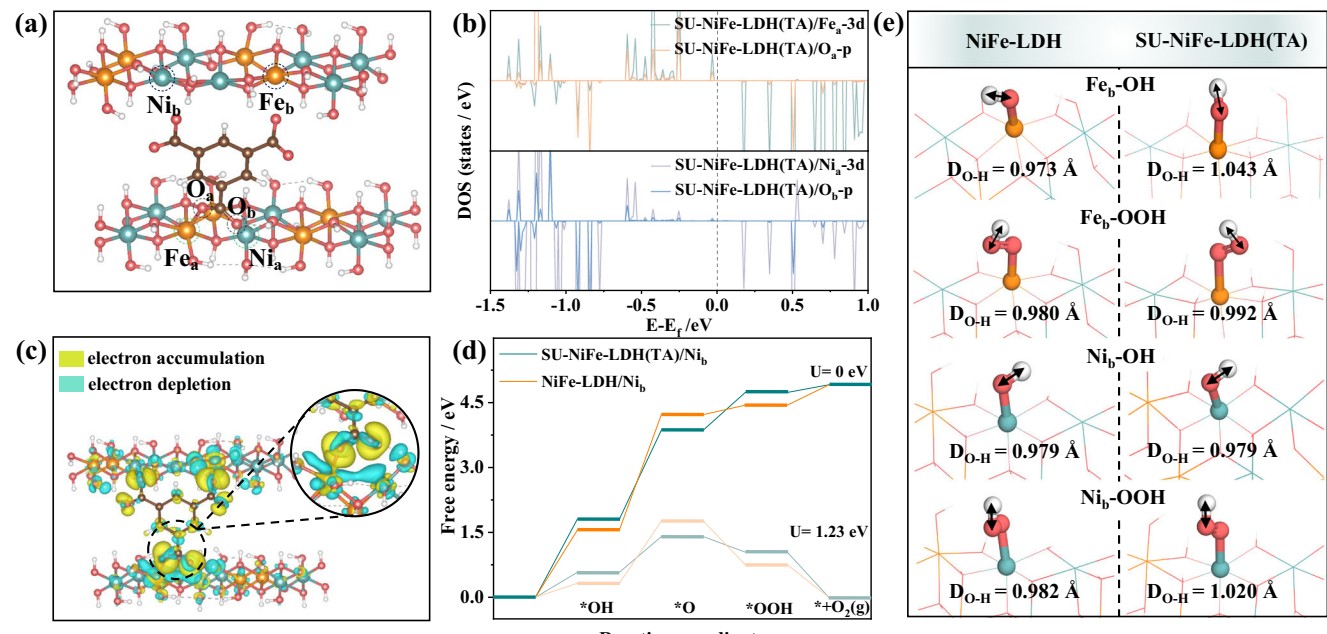

**Fig. 6 | Stable and activate mechanisms of functionalized ligands. a** Model for SU-NiFe-LDH(TA). **b** DOS of SU-NiFe-LDH(TA) between different sites, including $Fe_a$ and $O_a$, $Ni_a$ and $O_b$. **c** Differential charge density distributions of SU-NiFe-LDH(TA) model. **d** Free energy diagrams of NiFe-LDH and SU-NiFe-LDH(TA) models for $Ni_b$

sites. **e** The O–H bond length in *OH and *OOH of NiFe-LDH and SU-NiFe-LDH(TA) models at different adsorption sites ($Ni_b$ and $Fe_b$). Green balls: Ni atoms; Orange balls: Fe atoms; Red balls: O atoms; Brown balls: C atoms; White atoms: H atoms.

1500 mA cm$^{-2}$ without significant degradation or morphological collapse. In situ characterization and control experiments indicated that the high performance and stability originated from dynamic evolution of the uncoordinated carboxyl groups anchored on the LDH. That is, the coordinated carboxylates anchored via C–O–Fe bonds stabilized the metal active sites. The uncoordinated carboxylates ferried protons between the OER intermediates and bases to improve the OER performance. This work uncovered the intrinsic mechanism by which carboxylate ligands stabilized and activated metal centers and highlights the potential for commercial application of carboxyl ligand-functionalized catalysts in high-current water electrolysis.

## Methods

### Preparation of SU-NiFe-LDH(TA)@cp

In a typical procedure, electrochemical deposition was carried out with a standard three-electrode system comprising a Pt electrode and an Ag/AgCl electrode (saturation KCl) as the counter electrode and reference electrode, respectively. A piece of carbon paper (cp) in 1 × 2 cm$^2$ was used as the working electrode. Before electrodeposition, the carbon paper was cleaned by sequential ultrasonication in acetone and dilute HCl for 15 min to remove impurities. It was rinsed with water and ethanol before being dried in an oven at 60 °C. Then, a piece of the carbon paper was transferred into the electrochemical deposition solution containing 3 mmol Fe(NO$_3$)$_3$·9H$_2$O, 4.5 mmol Ni(NO$_3$)$_2$·6H$_2$O, 0.6 mmol trimesic acid, 20 mL of deionized water and 10 mL of DMF. The catalyst was prepared by the constant-voltage method at −1.0 V. High-quality ultrathin NiFe layered double hydroxide nanosheets modified by trimesic acid ligands (SU-NiFe-LDH(TA)@cp) were obtained after 1000 s of electrochemical deposition. After deposition, the SU-NiFe-LDH(TA)@cp electrode was rinsed with ethanol and deionized water and then dried in air at 70 °C. The active sites were detected by preparing control groups with varying ratios of Ni, Fe, and trimesic acid in the precursor. These control groups are referred to as SU-Ni$_x$Fe$_y$-LDH(TA$_z$)@cp, where $x$, $y$, and $z$ represent the molar quantities. The effective electrodeposition area was 1 × 1 cm$^2$. The catalyst loading was 3.0 ± 0.2 mg cm$^{-2}$.

### Preparation of NiFe-LDH@cp and NiFe-LDH(TA)@cp

The NiFe-layered double hydroxide nanosheet array (NiFe-LDH@cp) was synthesized by a procedure similar to that for SU-NiFe-LDH(TA)@cp. The electrochemical deposition solution contained 3 mmol FeSO$_4$·7H$_2$O, 4.5 mmol Ni(NO$_3$)$_2$·6H$_2$O, 20 mL of deionized water and 10 mL of DMF. The NiFe-LDH nanosheets modified by trimesic acid ligands were prepared by adding an additional 0.6 mmol of trimesic acid to the NiFe-LDH@cp preparation.

**Characterization.** The crystalline structures of the fabricated samples were characterized by X-ray diffraction (XRD, Philips X-Pert diffractometer) with Cu Kα radiation. The morphologies and structures were investigated with field-emission scanning electron microscopy (FE-SEM, Hitachi S-480 equipped with an energy dispersive X-ray spectrometer) and high-resolution transmission electron microscopy (HRTEM, JEM-2100F microscope with Cs-corrector). The surface compositions were investigated with X-ray photoelectron spectroscopy (XPS, a Thermo Fisher K-alpha 250Xi). Fourier transform infrared spectra (FTIR) were obtained with a Shimadzu IRTracer-100 using the attenuated total reflection (ATR) infrared mode. The dissolved metal ions in the OER electrolyte were measured by inductively coupled plasma–mass spectrometry (ICP–MS, Agilent 7700E). Raman spectra were measured with a DXR microscope (HORIBA HR Evolution). The contact angles for ultrapure water on the as-prepared electrodes were determined with a dynamic contact angle system (JC2000C1) at room temperature. Bubble adhesion and desorption at the surfaces of the composite electrodes were observed with an Axioplan 2 imaging system coupled with a CHI 660E electrochemistry workstation.

**Electrochemical measurements.** Electrochemical measurements were performed with a CHI 660E electrochemistry workstation using an alkaline medium (1 M KOH) and a traditional three-electrode configuration. Electrochemical measurements at large current densities were carried out with a workstation and a KA3005P programmable DC power supply. Linear sweep voltammetry (LSV) was performed at a

scan rate of 5 mV s⁻¹ with 90% iR compensation unless otherwise stated. All electrochemical curves are the result of repeated tests. The slopes of the curves were calculated with the Tafel equation:

$$\eta = b \times \log(j/j_0) \tag{1}$$

Electrochemical impedance spectroscopy (EIS) was conducted over a frequency range of $10^5$–0.01 Hz at 1.48 V vs. RHE. The CV curves were obtained over the range of 1.183 to 1.233 V vs. RHE. The ECSAs were determined from the double-layer capacitance ($C_{dl}$). The potentials described in this work were relative to the reversible hydrogen electrode (RHE), according to the Nernst equation:

$$E(RHE) = E(Ag/AgCl) + 0.0591 \times pH + 0.197 \tag{2}$$

**In situ FTIR Measurements.** The in situ FTIR device was assembled with SU-NiFe-LDH(TA)@nf as the anode and a steel disc as the cathode (denoted as SU-NiFe-LDH(TA)@nf$^{(+)}$//steel disc$^{(-)}$) to explore the evolution of the anode (Fig. S25a). To confirm the potential range, the water-splitting performance was tested, as shown in Fig. S25b. Based on the voltage window (1–3 V), the in situ FTIR spectra of SU-NiFe-LDH(TA)@nf were collected at different stages with intervals of 20 mV.

**Density functional theory (DFT) calculations.** All calculations were carried out with spin-polarized density functional theory (DFT) as implemented in the Vienna Ab initio Simulation Package (VASP) 6.1.0[38] with the Perdew-Burke-Ernzerhof (PBE)[39] generalized gradient approximation (GGA). The cutoff energy was set as 420 eV after cutoff testing, and the k-points were $2 \times 2 \times 1$ for the geometry optimizations. The k-points for electronic density of states calculations were set as $11 \times 11 \times 1$. In the densities of states analyses, the 3d orbitals were calculated for the Ni and Fe atoms, while the 2p orbitals of O were calculated for two O atoms in trimesic acid. The electronic energy and forces converged to $10^{-5}$ eV and 0.02 eV/Å, respectively. The van der Waals interactions were considered by the method of Grimme (DFT + D3).

Changes in the Gibbs free energies were calculated with the computational hydrogen electrode (CHE) model[40], in which the reaction H⁺(aq) + e⁻ = 1/2 H₂(g) was equilibrated at 0 V vs. the reversible hydrogen electrode at all pH values. The change in the Gibbs free energy (ΔG) for each elementary step was defined as follows:

$$\Delta G = \Delta E + \Delta E_{ZPE} - T\Delta S + \Delta G_U \tag{3}$$

where $\Delta E$, $\Delta E_{ZPE}$ and $\Delta S$ are the reaction energy, the zero-point energy (ZPE) and the entropy difference between the products and the reactants at room temperature (T = 298.15 K), respectively. $\Delta G_U$ is the contribution of the applied electrode potential (U) to ΔG, which was set as 0 V in our work.

## Data availability

The data generated in this study are provided in the Source data file and its Supplementary Information. The data used to generate the figures can be accessed in figshare (https://doi.org/10.6084/m9.figshare.23599278). Additional data are available from the corresponding authors upon reasonable request. Source data are provided with this paper.

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

## Acknowledgements

This work was supported by the National Natural Science Foundation of China (22101300), the Shandong Natural Science Foundation (ZR2020ME053, ZR2020QB027, ZR2022ME105 and ZR2023ME004), Qingdao Natural Science Foundation (23-2-1-232-zyyd-jch), State Key Laboratory of Enhanced Oil Recovery of Open Fund Funded Project (2022-KFKT-28), Major Special Projects of CNPC (2021ZZ01-05), and the Fundamental Research Funds for the Central Universities (22CX03010A, 20CX06007A and 22CX01002A-1) and the Entrepreneurship Practice Project of China University of Petroleum (202203007).

## Author contributions

X.L. (Xiaoqing Lu) led the project. X.L. (Xiaojing Lin) carried out the experiments and wrote the manuscript. X.C. repeated the experiments. X.Z., and H.C. finished the in-situ measurements. H.X., Q.H., D.S., and Z.C. proceed with the characterizations. S.W., S.C., and Y.H. conducted the density functional theory calculation. S.L., and Z.W. wrote the manuscript. All authors commented on the manuscript.

## Competing interests

The authors declare no competing interests.
