## [Peer Review File · Nature Communications]

REVIEWER COMMENTS

Reviewer #1 (Remarks to the Author):

This paper proposed a strategy to boost the OER performance of NiFe LDH by anchoring trimesic acid into the lamellar structure. The prepared SU-NiFe-LDH(TA) catalyst shows outstanding OER activity (248 mV@100 mA cm⁻²) and ultrastability at large current (1300 h at 1500 mA cm⁻²). Further comments are noted below:

1. In Figure S6, the peaks located at 1384 and 1501 cm⁻¹ should be contributed to vas(COO⁻) and aromatic CH groups, respectively. The characteristic infrared absorption wavenumbers of the uncoordinated (-COOH) carboxylate groups are normally strong in intensity and found in 1680-1800 cm⁻¹. Therefore, the FTIR spectrum of SU-NiFe-LDH(TA)@cp samples maybe has no uncoordinated (-COOH) carboxylate groups. Similarly, the coordinated carboxylate group presented a new doublet at ~ 1420 and ~1630 in Raman test, which were associated with the in- and out-of-phase stretching modes of the carboxylate group. Thus, it is difficult to distinguish the uncoordinated (-COOH) and coordinated (-COO⁻) carboxylate groups by Raman. In-situ FTIR seems more suitable for this work.
2. To gain further insight into the boosted OER performance, current density of LSV curves (Fig.4a) should be normalized by ECSA to compare the intrinsic activity for OER.
3. A surface restructuring process of NiFe LDH (crystalline state → amorphous state → (NiFe)OOH) is recognized on account of the published works about NiFe LDH and other (hydro)oxides in alkaline OER. Could the trimesic acid ligand endow the SU-NiFe-LDH(TA)@cp samples with no change in phase structure? Or whether the trimesic acid ligand was coordinated with (NiFe)OOH? Further characterization of the SU-NiFe-LDH(TA)@cp electrode after OER stability test should be conducted, including XRD, FTIR and Raman.
4. The abbreviation of NiFe-LDH@cp is hard for readers to follow on page 4, line 93. What is the cp? It is suggested to provide corresponding annotations.
5. Some NiFe LDH-related articles have identified the deactivation mechanism of NiFe LDH in OER (Adv. Mater. 2019, 31, 1903909; Angew. Chem. Int. Ed. 2021, 60, 24612–24619; Angew. Chem. Int. Ed. 2021, 60, 26829–26836), so the part 2.1 is unnecessary to be in the maintext.

6. Careful examination is needed to avoid typos and other problems (e.g. Figure 13a on page 8). Please double-check.

Reviewer #2 (Remarks to the Author):

Actually, the leaching of transition metals always happened in layered double hydroxides upon long-term stability test, which largely limits their activity at high current densities and large-scale commercialization. This manuscript submitted by Lu et al. adopted a static and dynamic compatibility anchor strategy to synthesize a subsize NiFe-LDH nanosheet catalyst (SU-NiFe-LDH(TA)@cp) via modifying with trimesic acid and regulating the type of electrodeposited ions. Benefited from the coordination between ligands and NiFe-LDH by C–O–Fe bonds and the promoted OER kinetic performance, the resulting superhydrophilic and superaerophobic SU-NiFe-LDH(TA)@cp catalyst exhibits enhanced electrocatalytic performance and stability at large current (1500 mA cm⁻²) for 1300 h. However, the general scope of this paper lacks innovation and is not of interest to a sufficiently broad audience. Over the past decade, LDH -based materials have been well studied in OER and it is not quite clear what is the new science in this work, especially combining the great tunability of enzymatic systems with known oxide-based catalysts to achieve both high activity and stability. Such as *Nat. Mater.* 2022, 21, 673–680; *ACS Appl. Mater. Interfaces* 2021, 13, 37063–37070. As thus, I would not recommend this manuscript to be published in *Nat. Commun.*

Other detailed comments:

1. If the authors want to prove that Fe³⁺ is conducive to the form of smaller nanosheet structure, you should supply the TEM and corresponding HRTEM images about NiFe-LDH(TA)@cp to reveal the structural distinction between SU-NiFe-LDH(TA)@cp and NiFe-LDH(TA)@cp according to the experimental details you mentioned.
2. Ni ion is the dominant part in NiFe-LDH, so why can't we see the C-O-Ni bonds in the spectra of Ni 2p for SU-NiFe-LDH(TA)@cp and NiFe-LDH(TA)@cp? Please give a reasonable explanation for the ligand's binding tendency to metal ions in LDH. Besides, the authors should list the XPS results of the three samples before and after OER test in tables and put them in SI, including the shift of peak position and the content of oxygen defect.
3. The authors emphasize that the coordinated carboxylates anchored on LDH by C–O–Fe bonds could stabilize the metal active sites to improve the OER stability and dynamics performance. So, in this NiFe system, which metal ion is the true catalytic active site or which metal ion dissolution is more detrimental to the stability of the structure? It is suggested that the authors should use XPS, XAS, etc to supply the relevant experimental data to strengthen the study of catalytic mechanism.

4. The authors should specify the electrochemical test conditions, especially the test voltage for electrolyzing water under the large current density (1500 mA cm^{-2}). Secondly, it is suggested that the author supply the test results under industrial conditions to highlight the excellent performance of the catalyst.

5. In fact, only the in situ Raman result of the SU-NiFe-LDH(TA)@cp is not enough to show that the coordinated carboxylates are favorable to stabilize the metal center and the involvement of carboxyl groups in proton transfer of OER intermediates. The authors should supply the in situ Raman data of other two samples (NiFe-LDH(TA)@cp and NiFe-LDH@cp) for comparative analysis. Moreover, It is recommended to supply the in situ infrared test to state the dynamic evolution behavior of carboxyl ligands during OER process.

6. There are many grammar and typo errors in this text. Thus, for a better readability, please check throughout the whole article text and correct all the errors.

Reviewer #3 (Remarks to the Author):

The work by X. Lin et al. reports a trimesic acid anchored NiFe-LDH with enhanced stability and activity for alkaline water oxidation. The authors argue that C-O-Fe bonds can be formed to stabilize LDH, thus realizing long-term stability as well as accelerated OER kinetics. Although the idea looks fancy, the major conclusions are not raised with solid evidence. Besides, some key characterizations, electrochemical results and DFT calculations seem to be conducted in an improper way. Considering these problems, I cannot suggest publication of this work. Below are some major concerns:

1. A key concern is that the stability tests of NiFe-LDH (Figure 1a) and NiFe-LDH(TA) (Figure S10) seem to be conducted under different potential. Such comparison cannot support the conclusion of enhanced stability.

2. Another key concern is that the evidence used to support the formation of C-O-Fe bonds is unconvincing. First, the FTIR analysis in Page 6 can only suggest the existence of trimesic acid and NiFe-LDH. The conclusion that 'trimesic acid ligand is successfully embedded in NiFe-LDH' seems to be speculation. Second, the XPS fitting in Figure 3 is far from rational. Taking Figure 3c as an example, the peak fitting seems to be conducted in a very subjective manner. Therefore, conclusions derived from Figure 3 are highly skeptical.

3. The measured XRD peaks in Figure S2 seem to be irrelevant to the labelled standard reference of Ni(OH)₂. The authors should clarify this point.

4. OER normally takes place at the uppermost surface layer of catalyst. However, based on the model in Figure 7a and Figure S21, the adsorption of OER intermediates are placed at the interlayer of LDH. The authors should explain more on this abnormal adsorption.

5. The charge density difference results in Figure 7c cannot tell how the charge redistributes at the interface. I suggest using Bader charge analysis.

6. Based on the proposed mechanism in Figure 6b, the hydrogen atom in OOH intermediate can interact with two oxygen atoms in trimesic acid. However, the OOH adsorption structure in Figure S20 and S21 cannot identify such a binding configuration. The authors should elucidate this inconsistency.

7. More details of the electrochemical tests should be provided, e.g., the scan rate, potential range and the potential set for stability test. The electrodes in this work were prepared on carbon paper. Therefore, the comparison data should also be those loaded on carbon paper. The authors should specify whether the comparisons in Figure 4c and 4g are those loaded on carbon paper.

Revisions and replies to the comments of the reviewers
(NCOMMS-23-05989)

REVIEWER COMMENTS

Reviewer #1 (Remarks to the Author):

This paper proposed a strategy to boost the OER performance of NiFe LDH by anchoring trimesic acid into the lamellar structure. The prepared SU-NiFe-LDH(TA) catalyst shows outstanding OER activity (248 mV@100 mA cm⁻²) and ultrastability at large current (1300 h at 1500 mA cm⁻²). Further comments are noted below:

Response: Thanks for your positive evaluation of our manuscript. We have revised the manuscript according to your valuable comments and suggestions.

1. In Figure S6, the peaks located at 1384 and 1501 cm⁻¹ should be contributed to vas(COO-) and aromatic CH groups, respectively. The characteristic infrared absorption wavenumbers of the uncoordinated (-COOH) carboxylate groups are normally strong in intensity and found in 1680-1800 cm⁻¹. Therefore, the FTIR spectrum of SU-NiFe-LDH(TA)@cp samples maybe has no uncoordinated (-COOH) carboxylate groups. Similarly, the coordinated carboxylate group presented a new doublet at ~ 1420 and ~1630 in Raman test, which were associated with the in- and out-of-phase stretching modes of the carboxylate group. Thus, it is difficult to distinguish the uncoordinated (-COOH) and coordinated (-COO-) carboxylate groups by Raman. In-situ FTIR seems more suitable for this work.

Response: Thanks for your valuable suggestion to improve our paper. We have substituted the prior FTIR spectrum with an attenuated total reflection infrared spectrum (Shimadzu IRTracer-100) to facilitate a precise characterization analysis of the material. For an intuitive comparison, the FTIR and Raman spectra of both trimesic acid (named as TA) and 1,3,5-Benzenetricarboxylic acid, sodium salt (named as TANA) are analyzed as well. Based on the results, the FTIR spectra are reorganized in **Figure**

S8. The spectral signals associated with M-OH bonds can be detected within the range of 400-650 cm^{-1} in layered double hydroxides. Compared with the results of NiFe-LDH, TA and TANA, the peaks at 1620 and 1433 cm^{-1} are ascribed to $\nu_{\text{as}}(-\text{COO}-)$ and $\nu_{\text{s}}(-\text{COO}-)$. And the broad bands at 769 and 727 cm^{-1} are related to the vibration of benzene ring in TANA. Significantly, two new peaks at 1160 and 850 cm^{-1} are observed in the SU-NiFe-LDH(TA) sample, which correspond to the C-O-Fe vibration modes. These results indicate that the trimesic acid ligand is successfully embedded in NiFe-LDH. The Raman spectra of TA and TANA indicate that the in- and out of phase stretching modes of the coordinated carboxylate groups for TANA are found in 1580 cm^{-1} and 1420 cm^{-1} , respectively. The broaden stretching modes at $\sim 1330 \text{ cm}^{-1}$ and $\sim 1646 \text{ cm}^{-1}$ are assigned to uncoordinated carboxylate group in TA (**Figure S9**).

Besides, the in situ FTIR tests are supplemented to unveil the dynamic evolution between uncoordinated (-COOH) and coordinated (-COO-) carboxylate groups for studying the electrochemical mechanism experimentally. The in situ FTIR device is assembled with SU-NiFe-LDH(TA)@nf and steel disc, which serve as the anode and cathode, respectively (denoted as SU-NiFe-LDH(TA)@nf⁽⁺⁾//steel disc⁽⁻⁾). In order to confirm the potential range, the water splitting performance (**Figure S25**) is test before the in situ FTIR test. Based on the voltage window (1-3 V), the in situ FTIR spectra of SU-NiFe-LDH(TA)@nf were collected in a voltage range of 1~3 V with an interval of 200 mV. The results were illustrated in **Figure 5b**. Besides the well-defined peak at 702 cm^{-1} attributed to the vibrations of M-OH is observed, another two peaks at 1620 cm^{-1} ($\nu_{\text{as}}(-\text{COO}-)$) and 1433 cm^{-1} ($\nu_{\text{s}}(-\text{COO}-)$) can be observed in the pristine SU-NiFe-LDH(TA)@nf at 0 V. With the increasing of voltage, two broaden stretching vibration modes at around 3000 cm^{-1} assigned to $\nu(-\text{OH})$ of uncoordinated carboxylate (-COOH) stand out and keep stable during the operation. Meanwhile, a new peak centered at 1608 cm^{-1} corresponding to the antisymmetric stretching vibration of -COOH ($\nu_{\text{as}}(-\text{COOH})$) gradually rises. These results further confirm the dynamic evolution of carboxyl ligands during water oxidation.

Figure S8. FTIR spectrum of NiFe-LDH@cp, SU-NiFe-LDH(TA)@cp, trimesic acid (TA) and 1,3,5-Benzenetricarboxylic acid, sodium salt (TANa).

Figure S9. Raman spectra of trimesic acid (TA), 1,3,5-Benzenetricarboxylic acid, sodium salt (TANa), NiFe-LDH@cp, NiFe-LDH(TA)@cp, and SU-NiFe-LDH(TA)@cp.

Figure 5. The in situ dynamic evolution of carboxyl ligand. a) In situ Raman spectroscopy measurements on SU-NiFe-LDH(TA)@cp sample at different applied potentials (V vs. Ag/AgCl). b) In situ FTIR spectroscopy measurements on SU-NiFe-LDH(TA)@cp sample at different applied potentials. c) Proposed OER pathways with intermediates for NiFe-LDH and SU-NiFe-LDH(TA).

2. To gain further insight into the boosted OER performance, current density of LSV curves (Fig.4a) should be normalized by ECSA to compare the intrinsic activity for OER.

Response: Thanks for your suggestion. The intrinsic catalytic activities of the catalysts were normalized by ECSA in **Figure S12**. It is clear that the SU-NiFe-LDH(TA)@cp catalyst shows the highest intrinsic activity among the samples, suggesting the increased activity results from not only the improvement of active surface area but also the introduction of trimesic acid ligand.

Figure S12. Intrinsic OER activity of the SU-NiFe-LDH(TA)@cp, NiFe-LDH(TA)@cp and NiFe-LDH@cp normalizing against ECSA.

3. A surface restructuring process of NiFe LDH (crystalline state \rightarrow amorphous state \rightarrow (NiFe)OOH) is recognized on account of the published works about NiFe LDH and other (hydro)oxides in alkaline OER. Could the trimesic acid ligand endow the SU-NiFe-LDH(TA)@cp samples with no change in phase structure? Or whether the trimesic acid ligand was coordinated with (NiFe)OOH? Further characterization of the SU-NiFe-LDH(TA)@cp electrode after OER stability test should be conducted, including XRD, FTIR and Raman.

Response: That is a good question. Indeed, the peak of M-OOH (M-O) can be observed at 529.4 eV in O 1s spectra for SU-NiFe-LDH(TA)@cp electrode after OER test (named as p-SU-NiFe-LDH(TA)@cp), which indicates that the surface restructuring of NiFe LDH is unavoidable in our OER process.

To investigate the coordination of trimesic acid ligand after phase restructuring, XRD, FTIR and Raman analyses are conducted on p-SU-NiFe-LDH(TA)@cp. As shown in **Figure S15**, the XRD pattern of p-SU-NiFe-LDH(TA)@cp is in accordance with that of the fresh SU-NiFe-LDH(TA). In **Figure S16**, the Raman characteristic peaks of p-SU-NiFe-LDH(TA)@cp shift to 474 and 544 cm^{-1} , respectively, which are

associated with the vibration mode of Ni-O in NiOOH. Besides, the Raman vibrations of coordinated and uncoordinated carboxylate are observed in p-SU-NiFe-LDH(TA)@cp, indicating that the trimesic acid ligand can be anchor to the NiFe-LDH stably during OER stability test. This perspective is further validated by the FTIR spectra of p-SU-NiFe-LDH(TA)@cp. As seen in **Figure S17**, the newly observed peak at 573 cm^{-1} can be attributed to the vibration of M-O. The signals associated with coordinated carboxylate can be detected at 1373 , 769 and 727 cm^{-1} . Those corresponding to the uncoordinated carboxylate can be detected at 1608 , 1454 , 1404 , 1276 , 1246 , 744 and 688 cm^{-1} . Besides, the vibration peaks of Fe-O-C are observed at 1160 and 850 cm^{-1} in the p-SU-NiFe-LDH(TA)@cp sample, which further qualifies the excellent structural stability.

Figure S15. XRD of SU-NiFe-LDH(TA)@cp and p-SU-NiFe-LDH(TA)@cp.

Figure S16. Raman spectra of SU-NiFe-LDH(TA)@cp and p-SU-NiFe-LDH(TA)@cp.

Figure S17. FTIR spectra of SU-NiFe-LDH(TA)@cp and p-SU-NiFe-LDH(TA)@cp.

4. The abbreviation of NiFe-LDH@cp is hard for readers to follow on page 4, line 93. What is the cp? It is suggested to provide corresponding annotations.

Response: We are sorry for this puzzling definition. The abbreviation of cp, cc, and nf stand for different conductive substrates: carbon paper, carbon cloth, and nickel foam, respectively, which has been explained in the revised manuscript.

5. Some NiFe LDH-related articles have identified the deactivation mechanism of NiFe

LDH in OER (Adv. Mater. 2019, 31, 1903909; Angew. Chem. Int. Ed. 2021, 60, 24612–24619; Angew. Chem. Int. Ed. 2021, 60, 26829–26836), so the part 2.1 is unnecessary to be in the maintext.

Response: We are appreciated for your helpful comments and we agree well with it. The part 2.1 has been moved to the Supporting Information. Appropriate changes in the main text have been made correspondingly, which influence little to the content and framework of the revised manuscript.

6. Careful examination is needed to avoid typos and other problems (e.g. Figure 13a on page 8). Please double-check.

Response: We are sorry for the mistakes such as typos. The manuscript has been carefully checked and revised by a native speaker to improve the English. The corresponding revisions have been highlighted in red in the main text.

Reviewer #2 (Remarks to the Author):

Actually, the leaching of transition metals always happened in layered double hydroxides upon long-term stability test, which largely limits their activity at high current densities and large-scale commercialization. This manuscript submitted by Lu et al. adopted a static and dynamic compatibility anchor strategy to synthesize a subsize NiFe-LDH nanosheet catalyst (SU-NiFe-LDH(TA)@cp) via modifying with trimesic acid and regulating the type of electrodeposited ions. Benefited from the coordination between ligands and NiFe-LDH by C-O-Fe bonds and the promoted OER kinetic performance, the resulting superhydrophilic and superaerophobic SU-NiFe-LDH(TA)@cp catalyst exhibits enhanced electrocatalytic performance and stability at large current (1500 mA cm^{-2}) for 1300 h. However, the general scope of this paper lacks innovation and is not of interest to a sufficiently broad audience. Over the past decade, LDH -based materials have been well studied in OER and it is not quite clear what is the new science in this work, especially combining the great tunability of enzymatic systems with known oxide-based catalysts to achieve both high activity and stability. Such as *Nat. Mater.* 2022, 21, 673–680; *ACS Appl. Mater. Interfaces* 2021, 13, 37063–37070. As thus, I would not recommend this manuscript to be published in *Nat. Commun.*

Response: Thanks for your great effort to improve our paper. Based on your valuable advice, we scrutinized our paper and made some changes. We have conducted a meticulous comparison and analysis of the articles you referenced. The innovations in the design of the hybrid electrocatalyst, the interaction mechanism and the catalytic effects are completely different. Yuan et al. reported a series of metal hydroxide-organic frameworks (MHOFs) combining the advantages of molecular catalysts and metal oxides through solvothermal reactions. The stability of MHOFs was governed by π -stacking interactions between linkers connecting adjacent hydroxide layers and the stability lasts for 20 h in 0.1 M KOH. (*Nat. Mater.* 2022, 21, 673-680) Yang et al. reported a NiFe layered double hydroxide intercalating a conductive polymer of

polypyrrole by anion exchange in salt-acid mixed solution, which improved the stability of the catalyst (20 mA cm^{-2} for 50 h) through the heightened electrostatic attraction. (*ACS Appl. Mater. Interfaces* 2021, 13, 37063-37070) Their research work and highlights have been discussed and cited in our manuscript. However, the synergy between the anchoring carboxylic molecules and the multi-metal active sites has never been fully understood in case of heterogeneous catalysis. The issues of phase segregation and active site reduction caused by metal dissolution in layered double hydroxides remain unresolved. The stability in the previous researches is far from reaching the industrial application standard as well. And, there is no direct evidence to uncover the mechanism of the stabilization and evolution of carboxylate ligands in catalysts. Therefore, our work differs from the previous work on the following points:

(1) Our work synthesized the subsize NiFe-LDH anchored with trimesic acid using electro-deposition method. Control experiments and theoretical calculations have revealed that the coordination between ligands and NiFe-LDH via C-O-Fe bonds optimizes the electronic structure, thereby inhibiting the dissolution of the bimetal active sites. The catalyst shows excellent stability under industrial conditions.

(2) The dynamic changes of carboxyl ligands during OER were visually detected by operando Raman and FTIR spectroscopy, which provided key evidence for the enhancement of kinetic. The uncoordinated carboxylates formed by dynamic evolution in OER process act as proton ferries to accelerate the OER kinetics.

The dynamic and static equilibrium of coordination and uncoordination carboxylates in water oxidation process is the key to maintain high activity and stability of catalysts. These new insights not only provide new insights to reveal the interaction between ligands and layered double hydroxide, but also enable the development of highly stable and efficient catalysts.

As such, we believe that our investigation is a good work with scientific meaning on energy storage field. And I would like to extend my sincere gratitude for your constructive comments once again. We have made extensive modifications to our

manuscript and supplemented extra data to make our results convincing.

Other detailed comments:

1. If the authors want to prove that Fe^{3+} is conducive to the form of smaller nanosheet structure, you should supply the TEM and corresponding HRTEM images about NiFe-LDH(TA)@cp to reveal the structural distinction between SU-NiFe-LDH(TA)@cp and NiFe-LDH(TA)@cp according to the experimental details you mentioned.

Response: We appreciate your comments and concur that the TEM and corresponding HRTEM images about NiFe-LDH(TA)@cp are necessary to elucidate the structure differences between SU-NiFe-LDH(TA)@cp and NiFe-LDH(TA)@cp. As depicted in **Figure S4b**, the HRTEM image of NiFe-LDH(TA)@cp reveals a similar interplanar spacing of 0.7 nm. Additionally, TEM images of SU-NiFe-LDH(TA)@cp and NiFe-LDH(TA)@cp are presented in **Figure S7**. It is evident that SU-NiFe-LDH(TA)@cp consists of smaller nanosheets (~56.68 nm) compared to NiFe-LDH(TA)@cp (~733.67 nm), which is consistent with SEM results.

Figure S4. HRTEM images of **a)** NiFe-LDH@cp, **b)** NiFe-LDH(TA)@cp, and **c)** SU-NiFe-LDH(TA)@cp, and the inset present SAED pattern of SU-NiFe-LDH(TA)@cp.

Figure S7. TEM image and the particle size distribution of **a)** SU-NiFe-LDH(TA)@cp and **b)** NiFe-LDH(TA)@cp.

The TEM and HRTEM images of NiFe-LDH(TA) have been supplemented in Figure S4b and Figure S7a, respectively.

2. Ni ion is the dominant part in NiFe-LDH, so why can't we see the C-O-Ni bonds in the spectra of Ni 2p for SU-NiFe-LDH(TA)@cp and NiFe-LDH(TA)@cp? Please give a reasonable explanation for the ligand's binding tendency to metal ions in LDH. Besides, the authors should list the XPS results of the three samples before and after OER test in tables and put them in SI, including the shift of peak position and the content of oxygen defect.

Response: Thanks for your valuable suggestion to improve our paper. Firstly, different from the distinguishable peaks of C-O-Fe bonds at 708.6 and 722.3 eV, the C-O-Ni peaks located at 856.7 and 874.8 eV are in close proximity to the those of Ni 2p_{3/2} and Ni 2p_{1/2} orbits, which are difficult to distinguish. (*Appl. Surf. Sci.*, 2022, 582, 152404) Meanwhile, the vibration peak of C-O-Ni bond is not observed in FTIR spectrum. Secondly, Fe exhibits a higher oxygen affinity than Ni. (*ACS Catal.* 2014, 4, 1, 289-

301; MIT Press: Cambridge, MA, 1971; pp 66-70) In this work, the results of DOS and charge density differences also indicate that the ligand is more likely to bind with Fe in LDH. As can be seen in **Figure 6b**, the 3d orbitals of SU-NiFe-LDH(TA)/Fe_a have more overlap and thus stronger interaction with 2p orbitals of SU-NiFe-LDH(TA)/O_a. Furthermore, the charge density difference (**Figure 6c**) and bader charge (**Figure S27**) analysis indicates that electrons can be transferred more easily from Fe to the ligand.

Besides, the comparison of XPS results for different electrocatalysts before and after OER test are listed in **Figure S18 and Table S3**. Particularly, the peaks at 856.1 (Ni 2p_{3/2}) and 873.7 eV (Ni 2p_{1/2}) in NiFe-LDH@cp increase to 856.8 and 874.4 eV ($\Delta E=0.7$ eV) after OER test (p-NiFe-LDH@cp). The Fe 2p peaks of p-NiFe-LDH@cp are located at 712.8 and 725.8 eV, which are 0.3 eV higher than those of NiFe-LDH@cp. Meanwhile, the ratio of O_v for p-NiFe-LDH@cp increases to 53.23%, indicating that the continuous metal dissolution creates more defects. However, few changes for the XPS spectra of NiFe-LDH(TA)@cp and SU-NiFe-LDH(TA)@cp can be observed after OER test (named as p-NiFe-LDH(TA)@cp and p-SU-NiFe-LDH(TA)@cp, respectively). It suggests that the anchoring strategy of carboxylate ligands is advantageous in stabilizing the electronic structure and coordination environment of the catalyst, thereby enhancing its stability.

Figure S18. High resolution XPS spectrum of **a)** Ni 2p, **b)** Fe 2p, **c)** C 1s, and **d)** O 1s for p-SU-NiFe-LDH(TA)@cp, p-NiFe-LDH(TA)@cp and p-NiFe-LDH@cp. The C=C bond originates from the substrate of carbon paper.

Table S3. Peak positions of Ni 2p and Fe 2p for the three samples before and after OER test and the relative content of oxygen defect (O_v) by XPS

Samples	Peak positions of Ni element		Peak positions of Fe element		O_v (%)
	Ni 2p _{3/2}	Ni 2p _{1/2}	Fe 2p _{3/2}	Fe 2p _{1/2}	
NiFe-LDH@cp	856.1	873.7	712.5	725.5	16.54
NiFe-LDH(TA)@cp	856.1	873.7	712.5	725.5	16.34
SU-NiFe-LDH(TA)@cp	855.5	873.1	712.5	725.5	26.43
p-NiFe-LDH@cp	856.8	874.4	712.8	725.8	53.23
p-NiFe-LDH(TA)@cp	856.1	873.7	712.5	725.5	16.38

REVIEWER COMMENTS

Reviewer #2 (Remarks to the Author):

We have carefully read the authors' responses and agree with the most of revisions they have made in response to our previous comments. However, there are still some questions that we think the authors need to further clarify, in particular those concerning comments 2 and 3. Therefore, I recommend publishing this manuscript after revisions. Below is a list of specific comments and questions the authors should address before publication.

1. NiOOH was generated on the catalyst surface according to the Raman results after OER stability testing and in situ Raman test results. However, there are no obvious shift of the peak position before and after OER test of the three samples listed in table S3. Please explain why high-priced nickel species are formed on the surface of the catalyst, but the valence state of the nickel does not change significantly for p-NiFe-LDH(TA)@cp?
2. Although the authors have demonstrated through a series of characterizations that the coupling between ligands and metal sites can significantly improve the stability of the catalyst, the source of the increased catalytic activity in this system has not been clearly explained. In fact, it is not enough to list the references showing that the dissolution of metal ions is the main source of catalyst deactivation. At the same time, the active site in the NiFe system has been controversial. We would like to know which metal or organic molecule is the source of activity in this work.

Reviewer #3 (Remarks to the Author):

I am glad that the authors have addressed some of my concerns and revised the manuscript accordingly. The quality of the revised version has been improved. However, before recommended for publication, the following issues need to be solved :

1. It's unconvincing to conclude that 'trimesic acid ligand is successfully embedded in NiFe-LDH'. The provided evidence seems to support surface-anchored trimesic acid rather than embedded trimesic acid. The authors should provide more direct evidence to justify this conclusion.
2. The authors need to elucidate why the valence state of Fe needs to be changed.
3. Kindly avoid using the word 'barrier' in the discussion of DFT calculations. Barrier refers to the transition state energy, which is not the case of this work.

Revisions and replies to the comments of the reviewers

(NCOMMS-23-05989A)

REVIEWER COMMENTS

Reviewer #2 (Remarks to the Author):

We have carefully read the authors' responses and agree with the most of revisions they have made in response to our previous comments. However, there are still some questions that we think the authors need to further clarify, in particular those concerning comments 2 and 3. Therefore, I recommend publishing this manuscript after revisions. Below is a list of specific comments and questions the authors should address before publication.

Response: Thanks for your positive evaluation of our manuscript. We have revised the manuscript according to your valuable comments and suggestions.

1. NiOOH was generated on the catalyst surface according to the Raman results after OER stability testing and in situ Raman test results. However, there are no obvious shift of the peak position before and after OER test of the three samples listed in table S3. Please explain why high-priced nickel species are formed on the surface of the catalyst, but the valence state of the nickel does not change significantly for p-NiFe-LDH(TA)@cp?

Response: Thanks for your valuable suggestion. The fitting results of XPS have been rechecked to ensure their accuracy. All the XPS peaks were calibrated by C 1s (284.8 eV) before comparison. As illustrated in **Figure R1**, compared to the shift of ~ 0.7 eV toward a higher binding energy for Ni 2p in bare NiFe-LDH@cp, no obvious change can be observed in both NiFe-LDH(TA)@cp and SU-NiFe-LDH(TA)@cp samples before and after OER test. The difference may be attributed to the dynamic evolution of trimeric acid. It is coordinated in the fresh SU-NiFe-LDH(TA)@cp while the uncoordinated one can be identified after the OER test. Protons in the uncoordinated trimeric acid may contribute to the redistribution of surface electrons, which perform simultaneously with the oxidation process. The similar Ni 2p shifts were observed in

NiCo LDH hybrid by Liu et al, where the peak of Ni²⁺ even shifted to lower binding energies after OER test, despite the inevitable surface oxidation. (*Angew. Chem. Int. Ed.* 2021, 60, 10614-10619)

Figure R1. The XPS results of Ni 2p for a) NiFe-LDH@cp and p-NiFe-LDH@cp, b) NiFe-LDH(TA)@cp and p-NiFe-LDH(TA)@cp, and c) SU-NiFe-LDH(TA)@cp and p-SU-NiFe-LDH(TA)@cp electrodes.

2. Although the authors have demonstrated through a series of characterizations that the coupling between ligands and metal sites can significantly improve the stability of the catalyst, the source of the increased catalytic activity in this system has not been clearly explained. In fact, it is not enough to list the references showing that the dissolution of metal ions is the main source of catalyst deactivation. At the same time, the active site in the NiFe system has been controversial. We would like to know which metal or organic molecule is the source of activity in this work.

Response: Thanks for your valuable suggestion to improve our paper. We strongly agree with you that the active site in the NiFe system is controversial. Recently, the generally accepted viewpoint on the main active site in NiFe-LDH is nickel, whereas the presence of Fe ions is essential to the high activity. (*Angew. Chem. Int. Ed.* 2020, 59, 8072-8077) Here, in order to figure out the active source, more samples with tunable compositions of metal ions and organic molecules were prepared. The OER performance can be further optimized by fine-tuning the dosage of precursors. The molar ratio of Ni and Fe precursors used in SU-NiFe-LDH(TA)@cp is 4.5:3 and the dosage of TA is 0.6 mmol. As shown in **Figure S20a** and **Table R1**, the activity of SU-Ni_{4.5}Fe_{3.5}-LDH(TA_{0.6})@cp decreased slightly when the dosage of Fe precursor was

increased with Ni/Fe ratio of 4.5/3.5 while it increased for SU-Ni₅Fe₃-LDH(TA_{0.6})@cp with increased dosage of Ni precursor (Ni/Fe ratio of 5/3). It suggests that Ni provides the major active site. As we know, trimeric acid ligands were successfully anchored on Fe sites by Fe-O-C bonds, resulting in more occupied Fe sites with more TA. However, the activity was improved by increasing TA content from 0.4 to 0.6 mmol (**Figure S20b** and **Table R2**). It demonstrates that Fe species may not be the primary active sites. Besides, by further increasing the content of trimeric acid in the precursor, the activity does not always increase with its content, ruling out the circumstance of organic TA directly serving as the active center. The coordinated carboxylates in ligands anchored to Fe and Ni species are essential to the high stability. And the uncoordinated carboxylates locate in the vicinity of the catalytic center and function as a ferry to capture and promote the proton transfer during OER.

Figure S20. Linear sweep voltammetry of samples (SU-Ni_xFe_y-LDH(TA_z)@cp) with different dosages of the precursors.

Table R1. Comparison of the activity between the SU-Ni_{4.5}Fe_{3.5}-LDH(TA_{0.6})@cp, SU-NiFe-LDH(TA)@cp, and SU-Ni₅Fe₃-LDH(TA_{0.6})@cp catalysts 1 M KOH electrolyte.

Samples	η at 10 mA cm ⁻² / mV	η at 100 mA cm ⁻² / mV
SU-Ni _{4.5} Fe _{3.5} -LDH(TA _{0.6})@cp	220	254
SU-NiFe-LDH(TA)@cp	220	248
SU-Ni ₅ Fe ₃ -LDH(TA _{0.6})@cp	216	243

Table R2. Comparison of the activity between the SU-Ni_{4.5}Fe₃-LDH(TA_{0.4})@cp, SU-NiFe-LDH(TA)@cp, and SU-Ni_{4.5}Fe₃-LDH(TA_{0.8})@cp catalysts 1 M KOH electrolyte.

Samples	η at 10 mA cm ⁻² / mV	η at 10 mA cm ⁻² / mV
SU-Ni _{4.5} Fe ₃ -LDH(TA _{0.4})@cp	224	270
SU-NiFe-LDH(TA)@cp	220	248
SU-Ni _{4.5} Fe ₃ -LDH(TA _{0.8})@cp	220	252

Reviewer #3 (Remarks to the Author):

I am glad that the authors have addressed some of my concerns and revised the manuscript accordingly. The quality of the revised version has been improved. However, before recommended for publication, the following issues need to be solved:

Response: Thanks for your great efforts. We have revised the manuscript carefully according to your valuable comments and suggestions.

1. It's unconvincing to conclude that 'trimesic acid ligand is successfully embedded in NiFe-LDH'. The provided evidence seems to support surface-anchored trimesic acid rather than embedded trimesic acid. The authors should provide more direct evidence to justify this conclusion.

Response: We apologize for not fully understanding this issue before. The description of the combination of organic and inorganic components has been revised as 'trimesic acid ligand is successfully anchored on NiFe-LDH'. On the one hand, during electro-deposition process the deprotonated trimeric acid binds to metal ions near the electrode. (*J. Am. Chem. Soc.*, 2011, 133, 12926-12929; *Nat. Commun.*, 2019, 10, 5074) Then, they react with the hydrated Ni²⁺ ions near the electrode, leading to the formation of LDH phase as well as anion intercalation and ultimately growth before aging of the LDH phase. (*ACS Catal.*, 2020, 10, 11127-11135; *Dalton Trans.*, 2013, 42, 15687-15698) On the other hand, the interlayer spacing of NiFe-LDH (~ 7.70 Å) is large enough to accommodate trimeric acid groups (7.02 Å). (*Nat. Commun.*, 2014, 5, 4477)

Therefore, trimesic acid groups are most likely anchored to metal ions on the surface of the layered structure. To avoid confusing the reader, the word 'embedded' has been replaced by 'anchored' in the revised manuscript.

2. The authors need to elucidate why the valence state of Fe needs to be changed.

Response: Thanks for your suggestion. The subsize NiFe-LDH nanosheets catalyst modified with trimesic acid (SU-NiFe-LDH(TA)@cp) could be achieved by using Fe³⁺ precursor. According to the research findings of Sun et al., the preparation of NiFe-LDH starts with the precipitation of ferric oxyhydroxides, FeO(OH), rather than the

Ni(OH)₂ due to the difference of the solubility. Then, the adsorption and nucleation of nickel species take place on the surface of FeO(OH) for generating Ni(OH)₂ phase. Subsequently, the iron on the surface of ferric oxyhydroxides would diffuse into the nearby Ni(OH)₂ phase and substitute some nickel sites, leading to the formation of LDH. (*ACS Catal.*, 2020, 10, 11127-11135) The pH of the precursor can directly affect the dynamic equilibrium of dissolution/hydrolytic precipitation during LDH formation. We further test the pH values of NiFe-LDH(TA) and SU-NiFe-LDH(TA) precursors by pH test pen (lichen, pH-100A). As shown in **Figure S7**, the pH value of the reaction solution containing Fe³⁺ is lower than that containing Fe²⁺, due to the small acidity coefficient of Fe³⁺ in comparison with that of Fe²⁺ (pKa[Fe³⁺]=2.84, pKa[Fe²⁺]=6.74). Observed from the SEM images in **Figure S8**, the size of SU-NiFe-LDH(TA)@cp nanosheet is reduced to 56.68 nm, compared to NiFe-LDH(TA)@cp prepared with Fe²⁺ (~ 733.67 nm). In **Figure 4**, the contact angle tests for water droplet and the release behavior of O₂ bubbles indicate that the smaller nanosheet is beneficial to the superhydrophilic and superaerophobic, which are conducive to improve activity and stability.

Figure S7. The pH value in the precursor solution of Fe²⁺ and Fe³⁺ for NiFe-LDH(TA) and SU-NiFe-LDH(TA), respectively.

3. Kindly avoid using the word ‘barrier’ in the discussion of DFT calculations. Barrier refers to the transition state energy, which is not the case of this work.

Response: We are appreciated for your helpful comments and sorry for this mistake.

The discussion in our paper has been revised as follows.

“The rate-determining step (RDS) of OER on the Ni_b and Fe_b sites in NiFe-LDH is respectively determined to be the formation of *O and *OOH, where the highest free energy changes were found to be 2.67 and 2.12 eV at U = 0 V.” (Line 2, Page 12)

REVIEWERS' COMMENTS

Reviewer #2 (Remarks to the Author):

Overall, I'm happy with the revision and would recommend to publish the revised version with any editorial corrections as needed.